# *Minjung* Theology as a Project of Profanation: Focusing on the *Minjung*-Event Theory of Byung-Mu Ahn

Yongtaek Jeong 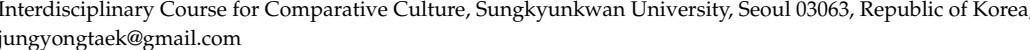

Interdisciplinary Course for Comparative Culture, Sungkyunkwan University, Seoul 03063, Republic of Korea; jungyongtaek@gmail.com

**Abstract:** The relationship between *minjung* theology and the process of social change called secularization or theoretical and practical projects based on such processes of social change is complex. It requires more detailed discussions. Therefore, this paper seeks to reinterpret *minjung* theology as a theological *minjung* project using the methodology of new-style phenomenology of religion with a theoretical basis on Italian philosopher Giorgio Agamben's conceptions of secularization and profanation as projects with religious intentions and orientations. Through this reinterpretation, the paper demonstrates that *minjung* theology in relation to secularization is a unique theological project with very different goals from those of Latin American liberation theology as well as other political and situation theologies. In order to accomplish this purpose, the paper first introduces French sociologist Émile Durkheim who has explained secularization differently from German sociologist Max Weber. It then shows that secularization is not the only way in which the sacred is reappropriated through Agamben's discussions of secularization and profanation. To identify the passage from secularization to profanation of the concept of *minjung*, this paper analyzes the *minjung*-event theory of Byung-Mu Ahn, a representative first-generation *minjung* theologian. This theory emphasizes the importance of "event" as a way of understanding *minjung* instead of defining it conceptually. Insofar as it presents the *minjung* as an intrinsically unnamable, invisible, and unpredictable event, a form of religious phenomenon called "the sacred", *minjung*-event theory involves an attempt to secularize Jesus-Messiah as the *Minjung*-Messiah. In conclusion, this paper argues that beyond the secularization of the Messiah into the *Minjung*, *minjung*-event theory moves toward a dialectical project of desacralization and re-sacralization, in which the *minjung* itself is profaned into an event.

**Keywords:** *minjung* theology; liberation theology; profanation; secularization; *minjung*-event; the sacred; the profane; *minjung* project; new-style phenomenology of religion; internal rupture of structure

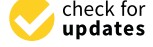



## 1. Introduction: Is *Minjung* Theology a Form of Liberation Theology?

> "Liberation theology and *Minjung* theology are different [. . .] the concept of *minjung* [. . .] encompasses a broader social and cultural reality."
>
> —Suh (1983, p. 228)

The journal *Religions* has recently published a series of three articles on the topic of *minjung* theology (*Minjungshinhak*).[1] First, there was Andrew Eunghi Kim's 2018 paper, which argued that "in light of liberation theology, including *minjung* theology", there was a "need to rethink both the role of religion in contemporary settings and the theory of secularization" (Andrew Eungi Kim 2018, p. 13). This was followed by Young Hoon Kim's 2020 paper, which analyzed the Korean novelist Hwang Sok-Yong's masterpiece, *The Guest*, "in relation to the emotional complex of han as understood in Korean minjung theology, the political theology of Johann Baptist Metz, and Ignacio Ellacuría's liberation theology" (Young Hoon Kim 2020, p. 1). Finally, the most recent article was Sam Han's 2021 paper, which attempted to establish a dialogue between *minjung* theology and contemporary political theory "by revisiting minjung theology's contribution to the understanding of

han as an emotional epistemology of subjugated social groups centered on relieving the conditions of ressentiment" (Sam Han 2021, p. 21).

Coincidentally, all three papers referred to *minjung* theology and introduced it as "liberation theology in the Korean context", namely "people's theology", or "the Korean version of 'liberation theology'" (Andrew Eungi Kim 2018, p. 1; Young Hoon Kim 2020, p. 5; Sam Han 2021, p. 1). All three authors identified Korean *minjung* theology as one of the offshoots of Latin American liberation theology.[2] In particular, Andrew Eungi Kim's paper, the first of the three to be published and referenced by both subsequent researchers Young Hoon Kim and Sam Han, not only presents *minjung* theology as one of "other forms of liberation theology", but also argues that *minjung* theology as liberation theology "rejects the decline of religion thesis of secularization theory" (Andrew Eungi Kim 2018, p. 3). According to him, "by lending theological support to various socio-political causes and by becoming more socially involved in the fight for justice", "liberation theology, including minjung theology" "manifests the changing role of religion that is more socially concerned and involved", insofar it shows "a greater involvement of Christianity in this-worldly matters on the side of the underprivileged" (Andrew Eungi Kim 2018, p. 10). In other words, if we redefine secularization as "the increased use of sacred institution—belief and practice—for secular purposes", then the development of liberation theology, including *minjung* theology, "directly disputes the decline of religion thesis", insofar "as underprivileged groups fight for justice in the name of God or gods" (Andrew Eungi Kim 2018, p. 10). In short, his argument is that the dominant thesis of secularization theory, which asserts the decline of religion, is no longer acceptable because religion has actually become more important to many people in the contemporary society, at least in terms of the development of liberation theology, which has struggled for justice through the logic of faith.

Andrew Eungi Kim's argument that *minjung* theology is a form of liberation theology, and the fact that liberation theology, including *minjung* theology, refutes the dominant thesis of secularization theory of the decline of religion, are compelling on their own terms once we set aside the question of whether *minjung* theology is indeed a form of liberation theology.

However, the relationship between *minjung* theology and the process of social change called secularization, or theoretical and practical project based on such process of social change, is complex. It requires more detailed discussions. If we define secularization as simply "the decline of religion," as Kim does, as "process by which sectors of society and culture are removed from the domination of religious institutions and symbols" (Berger 1967, p. 107), then, in the case of Byung-Mu Ahn, the representative first-generation *minjung* theologian whom this paper focuses on, there is no doubt that he not only recognizes secularization as an ongoing and inevitable phenomenon that is universal in world history, but also accepts it as a task to be actively taken up for the development of *minjung* theology. For example, when Ahn called for the "de-Westernization of Christianity," saying that "we no longer need to meet Jesus through the medium of the West, nor do we need to express our feelings in their tune and style. De-Westernization! This will be the way to narrow the distance between Jesus and us" (Ahn 1999, p. 430), his call for the "de-Westernization of Christianity" was nothing less than an active practice of secularization in the sense of liberation from formal aspects of religion symbolized by Western Christianity and Western theology, especially its institutional authority.[3] Thus, if, contrary to Andrew Eungi Kim's claim, *minjung* theologians actually sought to actively practice secularization, the relationship between secularization and *minjung* theology also requires more careful consideration.

Based on this problematic, this paper seeks to reinterpret *minjung* theology as a theological *minjung* project using the methodology of new-style phenomenology of religion, with a theoretical basis on Italian philosopher Giorgio Agamben's conception of secularization and profanation as projects with religious intentions and orientations. Through this reinterpretation, the paper demonstrates that *minjung* theology in relation to secularization

is a unique theological project with very different goals from those of Latin American liberation theology as well as other political and situation theologies.

To accomplish this purpose, in Section 2, this paper first introduces French sociologist Émile Durkheim, who has explained secularization differently from German sociologist Max Weber. It then shows that secularization is not the only way in which the sacred is reappropriated through Agamben's formulations about the difference between secularization and profanation. In this article, the strict conceptual distinction between secularization and profanation, based on Agamben's philosophy, is important not only for interpreting the religious intentions and orientation of *minjung* theology, but also for distinguishing it from liberation theology, which is discussed in detail in Section 2. To identify the passage from secularization to profanation of the concept of *minjung*, in Section 3, this paper analyzes Byung-Mu Ahn's *minjung*-event theory, which emphasizes the importance of "event" as a way of understanding *minjung* instead of defining it conceptually. Here, the paper undertakes the task of reformulating *minjung*-event theory as a project of profanation. Insofar as it presents *minjung* as an intrinsically unnamable, invisible, and unpredictable event, a form of religious phenomenon called "the sacred", *minjung*-event theory involves an attempt to secularize Jesus-Messiah as the *Minjung*-Messiah. Section 4 explains the ways in which *minjung* theology's *minjung*-event theory overcomes some epistemological limitations shared by Korean *minjung* theories in general in the 1970s and 1980s. In its conclusion, this paper argues that beyond secularization of the Messiah into the *Minjung*, *minjung*-event theory moves toward a dialectical project of desacralization and re-sacralization which profanes *minjung* itself into an event.

## 2. Revisiting Concepts of Secularization and Profanation: Based on Discussions of Durkheim and Agamben

> "In all probability, the concepts of totality, society, and deity are at bottom merely different aspects of the same notion."
>
> —Durkheim ([1912] 1995, p. 443)

As Andrew Eungi Kim himself notes, "the concept of secularization is multidimensional, entailing many different meanings" (Andrew Eungi Kim 2018, p. 8). He notes that among various meanings of secularization, "five points are most representative: decline of religion; social change; institutional differentiation; rationalization; and privatization". He confirms that "the most familiar notion of secularization is the decline of religion" (Andrew Eungi Kim 2018, p. 8) and emphasizes that "secularization as a process of social change refers to a shift from 'sacred' to 'secular' society", after all (Andrew Eungi Kim 2018, p. 9).[4] In particular, when he considers that "the main problem with secularization theory is its many unwarranted assumptions, particularly that of relating the concept with the decline of religion" (Andrew Eungi Kim 2018, p. 9), it becomes clear that his thorough understanding is based on a Weberian model of secularization, which understands secularization as the product of a rationalization process that is "disenchantment of the world" (Weber [1919] 2009, p. 155).

However, the Weberian model is not the only one in the secularization theory. While Weber's theory of secularization is based on the conviction that "by rationalization and intellectualization and, above all, by the 'disenchantment of the world' . . . the ultimate and most sublime values have retreated from public life either into the transcendental realm of mystic life or into the brotherliness of direct and personal human relations" (Weber [1919] 2009, p. 155), another classical sociologist Durkheim has a very different understanding of the process of secularization than Weber's, as he argues that "now as in the past, we see that society never stops creating new sacred things" (Durkheim [1912] 1995, p. 215). In other words, while Weber posits that due to secularization, religious charisma becomes increasingly routinized, Durkheim argues that, concentrated, the sacred has expanded to peripheral areas. Thus, although the secular has been strengthened in opposition to the religious in modern society, Durkheim maintains that the sacred, contrasting with the profane, still persists, because, even with the strong force of secularization in modern

society, there remains a need to distinguish between the sacred and the profane (Jong-Ryul Choi 2006, p. 347). Consequently, Durkheim contends that despite ongoing social changes accompanied by modernization, each society remains unified because of "a unified system of beliefs and practices relative to sacred things, that is to say things set apart and forbidden", or, in short, "religion" envelopes society as a whole (Durkheim [1912] 1995, p. 44; cf. Fenn 1978, p. xiii; Chongsuh Kim 2005, p. 132).

Political theologian Carl Schmitt, quite similarly to Durkheim, opposes Weber's understanding of secularization and offers a completely different interpretation. According to the Italian philosopher Giorgio Agamben, "while, for Weber, secularization was an aspect of the growing process of disenchantment and detheologization of the modern world, for Schmitt it shows on the contrary that, in modernity, theology continues to be present and active in an eminent way" (Agamben 2011, pp. 3–4). According to Weber's definition, advancement of secularization means gradual disappearance of the religious or the theological in the public sphere. In contrast, Schmitt argues that secularization is really just Protestant theology that has moved underground. Schmitt's famous thesis, "all significant concepts of the modern theory of the state are secularized theological concepts", encapsulates his understanding of secularization (Schmitt 2005, p. 36).[5]

Agamben follows an alternative definition of secularization advanced by Schmitt. For Schmitt and Agamben, theology is still present at the core of our conceptions of the state. To explain how this works, Agamben calls secularization a "signature": "something that in a sign or concept marks and exceeds such a sign or concept referring it back to a determinate interpretation or field, without for this reason leaving the semiotic to constitute a new meaning or a new concept." In other words, they "move and displace concepts and signs from one field to another (in this case, from sacred to profane, and vice versa) without redefining them semantically" (Agamben 2011, p. 4). However, secularization is not the only way in which the sacred is reappropriated. The sacred can also be "profanized" without being secularized. For Agamben, the distinction between secularization and profanation is crucial in this respect. Agamben explains the difference between secularization and profanation as follows:

> Secularization is a form of repression. It leaves intact the forces it deals with by simply moving them from one place to another. Thus the political secularization of theological concepts (the transcendence of God as a paradigm of sovereign power) does nothing but displace the heavenly monarchy onto an earthly monarchy, leaving its power intact. Profanation, however, neutralizes what it profanes. Once profaned, that which was unavailable and separate loses its aura and is returned to use. Both are political operations: the first guarantees the exercise of power by carrying it back to a sacred model; the second deactivates the apparatuses of power and returns to common use the spaces that power had seized. (Agamben 2007, p. 77)

Profanation, which means returning something to a free and human use, should not be confused with secularization. If the distinction between the sacred and the profane does not depend on material attributes of the object, place, animal, or person in question, then any object considered sacred can be reverted to the profane realm at any time. In essence, if the distinction between the sacred and the profane lies in the act of separation itself, practically all objects can be made sacred or, conversely, profane. What matters is how an object is used. Agamben argues that an object separated from common use through sacralization can be returned to common use through profanation (Agamben 2007, p. 82). However, it is crucial to emphasize that profanation does not destroy the object in question nor reverts it to its presumed natural use before its separation into the realm of the sacred. Instead of destroying the sacred use of an object, profanation deactivates or disables it, making it available for new uses (Phelps 2014, pp. 639–40; Prozorov 2014, pp. 43–44).

Profaning a sacred object shifts it from a distinct realm, where its application is restricted or controlled, to one with open accessibility. This transformation allows the free usage of the object in numerous unconventional manners. As a result, the object becomes a

"pure means"—separated from any designated purpose and defined solely by its inherent potential for diverse uses. As Agamben says, profanation is the process of shifting an object into "the sphere of those means that emancipate themselves from their relation to an end while still remaining means" and the experiences and phenomena that accompany that process (Agamben 2000, p. ix; Prozorov 2011, pp. 77–78). For this reason, the goal of profanation is "not simply to abolish and erase separations but to learn to put them to a new use, to play with them" (Agamben 2007, p. 87). In other words, to profane is not to abolish all distinctions between the sacred and the profane, but rather to transform differences between the sacred and the profane into pure means by disabling the apparatuses of power that create those distinctions in the first place in order to create new possibilities for the use of what was designated as sacred and what was designated as profane.

In terms of re-appropriation of the sacred, secularization operates on the premise that the sacred and the profane are incompatible and mutually exclusive. On one hand, it is a process that renders everything profane. Thus, discerning the difference between the sacred and the profane is impossible. On the other hand, there is a universal endeavor in every sphere of social life to rediscover the meaning of life through encounters with an indelible sacrality, which, while allowing sociality, can never be fully eradicated. After all, secularization as a modern project to regulate/control the sacred can be understood as a process that leaves only the cognitive dimension in the social world while attempting to exclude emotional and ethical dimensions. This is part of an official project to desacralize or disenchant, behind which the pursuit of sacrality persists in fragmented forms (resacralization). Therefore, the project of secularization is inevitably intertwined with the will to power, aiming to regulate/control sacrality. It is based on a binary symbolic classification of the sacred and the profane that separates the everyday profane world from its opposing sacred world. On the other hand, in the project of profanation, "it is not a matter of rejecting the sacred sphere but of identifying the points at which the two clearly fail to separate in order to deactivate the separation itself" (Murray 2010, p. 126).

To reinterpret more actively, profanation acknowledges that "the sacred paradoxically coexists with the profane in every religious manifestation" since "anything is potentially hierophanic" (Allen 1998, p. 92). It can be described as a process where "the sacred continually historicizes and limits itself in new objects and assumes new forms while at the same time attempting to disengage itself and realize its essential structure; and the sacred, in revealing itself, conceals and hides itself" (Allen 1998, p. 92). In Agamben's words, to enact profanation is to "put the distinction between sacred and profane into crisis", insofar as "the religious machine seems to reach a limit point or zone of undecidability, where the divine sphere is always in the process of collapsing into the human sphere and man always already passes over into the divine" (Agamben 2007, p. 79). In fact, this is a project aimed at discovering "a residue of profanity in every consecrated thing and a remnant of sacredness in every profaned object" (Agamben 2007, p. 78). Through this project of profanation, one can eventually grasp "how the sacred is able to manifest itself in its complexity, ambiguity, and profundity" and "it becomes possible to gain some appreciation of the immensity of that which is hidden behind any disclosure and of its hidden potential as an inexhaustible source of creativity and meaning" (Allen 1998, p. 92).

Such a reinterpretation of secularization and profanation suggests that in order to capture the manifestation of the sacred through secularization or profanation, it is insufficient to merely describe the religious phenomena transforming in contemporary society. Instead, we must understand them within more complicated and multilayered subjective intentions and orientations of those participating in religious activities, which cannot simply be reduced to theology. Particularly, following the methodology of the New Style Phenomenology of Religion,[6] which argues for treating intentions and orientations of contemporary religious individuals and their communities as factual data for empirical research on religion, secularization and profanation should be viewed as "not religious phenomena in and of themselves, but the religious and other interpretations and applications

given to them, which should be the real object of research" (Waardenburg 2001, p. 449). Of course, these demands can equally be applied to the study of *minjung* theology.

As Korean researcher Namhee Lee argued and as sociologist of religion In-Cheol Kang who recently published a monograph on *minjung* theory affirmed, in contemporary Korean society, *minjung* is not just a concept, but functions as the "movement" (*minjung* movement) and the "*minjung* project" (Namhee Lee 2007, p. 1; In-Cheol Kang 2023a, p. 16). In other words, *minjung* is "a product of the complex interplay of structural preconditions, Korea's repressive military regimes and its concomitant rapid industrialization, and the *minjung* movement's own 'political culture'" and "discursive contestations in a field of political, cultural, and symbolic forces" (Namhee Lee 2007, pp. 1–2). Therefore, *minjung* theology could also be understood as a historically unique theological "*minjung* project" that emerged particularly within the Korean ecumenical social mission and the Protestant democratization movement camp in the 1970s and 80s, among various attempts made in different sectors (academics, arts, religion, etc.) to actualize the concept of the "*minjung*". Hence, the thesis of secularization and profanation raised in the debate on the relationship between modernity and religion could also be interpreted religiously from the perspective of intentions and orientations of religious actors who practiced the theological *minjung* project, referring to the new style phenomenology of religion's approach to "the sacred" as a social reality.

### 3. *Minjung* as the Event Itself: An Overview of Byung-Mu Ahn's *Minjung*-Event Theory

> "*Minjung* theology is not a discipline that treats *minjung* as an object. Instead, it's about verbalizing the experiences of events brought about by *minjung* as the subject and taking on the role of a witness."
>
> —Byung-Mu Ahn (1993b, p. 256)

In Korea, the term "*minjung*" first emerged during the Japanese colonial era in the 1920s and 1930s. It continued to be used after the 1945 Liberation of Korea. However, until 1960s, the concept of *minjung* primarily remained within the language of social movements. It was not until the early 1970s with the establishment of the authoritarian regime under Chung-Hee Park that the concept of *minjung*, which had virtually disappeared since the late 1930s and took nearly 40 years to re-emerge as a resistance political entity, became academically integrated (Sangchul Jang 2007; Namhee Lee 2007). Only then did various *minjung*-oriented approaches appear in various academic disciplines, such as "*minjung* literature", '*minjung* theology", "*minjung* economics", "*minjung* sociology", "*minjung* education", and "*minjung* history". These discourses from each discipline collectively led to the emergence of "*minjung* studies", a multi-disciplinary and inter-disciplinary field in its initial form known as "*minjung* theory" (In-Cheol Kang 2020, p. 247). Therefore, it could be said that "the 1970s was the period when *minjung* began to be used for the first time as the 'language of social movement and politics' and as 'academic terminology' or 'language of the academic community'" (In-Cheol Kang 2023a, p. 49). By the 1970s, "*minjung* were no longer evaluated as passive and dependent beings guided and enlightened by intellectuals or elites, but as proactive, independent, active, creative beings with abundant transformative potential" (In-Cheol Kang 2023b, pp. 165–66).

In the 1970s, *minjung* theologians developed their theological discourses on *minjung* through dialogues with scholars from other disciplines who discussed *minjung*. In other words, the theological conceptualization of *minjung* in *minjung* theology would not have been possible without engaging in discussions with contemporary humanities and social science discourses on *minjung*, such as the "Idea of Ssial" (Seok-Heon Ham), "*Minjung* Poetry" (Ji-Ha Kim), "*Minjung* Literature" (Nak-Cheong Baek), "*Minjung* Economics" (Hyun-Chae Park), "*Minjung* Sociology" (Wan-Sang Han), and "*Minjung* History" (Man-Gil Kang). Thus, when *minjung* theologians were developing their theology, academic discourses on *minjung* in Korea served as a significant reference. In many instances, commonalities and differences were identified. In this regard, In-Cheol Kang asserted that

the conception of "resistant *minjung*", which largely persisted into the 1980s and 1990s, was first established in the 1970s. He evaluated that the "resistant *minjung* concept that re-emerged in the 1970s encompassed the elements of the majority, the oppressed, hierarchical multidimensionality, subjectivity (subject of history and political subject)" (In-Cheol Kang 2023b, p. 165). Indeed, *minjung* theology also shared this resistant *minjung* concept, which included these four elements, with other *minjung* theories. However, as years progressed through the 1990s and into the 2000s, *minjung* theories of the 1970s and 1980s faced criticism. Similarly, in *minjung* theology, criticisms were also raised against earlier generations by the third-generation *minjung* theologians who emerged in the mid-1990s (see, for a typical example, Jin-Ho Kim 2013, pp. 203–5).

First, the epistemological basis of the 1970s and 80s *minjung* theory, known as the "scientific and transformative *minjung* theory", had a limitation of normatively and statically setting *minjung* as a singular transforming subject (Yong-Ki Lee 2010b, p. 5). That is, it held the common view that *minjung* was a unified and essential entity molded through dialectics of national contradictions and class contradictions and seen as a teleological subject ultimately moving towards self-liberation (Yong-Ki Lee 2010a, p. 12). Even if, like *minjung* theologians, the majority of *minjung* theorists understood *minjung* not as a unified organizational entity with a well-defined internal order, but as a collective of subjects with diverse identities, *minjung* of that era was represented not as a fluid and constructive being formed, dismantled, and reorganized by various human groups within historically specific conjunctures, but as a singular Subject (upper case subject) striving towards a singular goal. The resistance of *minjung* against the existing order tended to strongly align with an optimistic and progressive (or developmental) view of history.

Second, on the surface, *minjung* were claimed to be proactive agents voluntarily carrying out the transformative movement. However, they were implicitly assumed to be passive entities that needed guidance by the elite (Yong-Ki Lee 2010a, p. 12). Especially in *minjung* theology, *minjung* were often declared as the "Subject of History". Yet, in reality, there was a strong tendency to objectify and marginalize them in historiography. Many *minjung* theologians exposed problems of reducing the subjectivity of *minjung* to some essence within them or directly equating external structural conditions with attributes of agents without any mediations by extracting common attributes from various subalterns or minorities that could be categorized as *minjung* (Yongtaek Jeong 2013, p. 166). When *minjung* were easily declared as the Subject of History, the complex and diverse temporality of history was reduced to "the becoming and self-production" of "the collective singular subject". Thus, the "empirical history" returned as "belonging to the *Minjung*" consequentially separated and alienated "the concrete individuals" referred to as *minjung* "from their essential species attributes", for instance, their "*minjung*-ness" (Benhabib 1986, p. 57).

Third, traditional *minjung* theory posits *minjung* based on a dichotomy of dominance and resistance, presenting them as a single resisting entity against a singular dominant force (Yong-Ki Lee 2010a, p. 12). Defined as the sole agent of transformational movement, the history of *minjung* and the history of *minjung* movement become synonymous, thereby severely diminishing the historical significance of the lives of *minjung* who do not resist and their daily lives outside moments of struggle (Yong-Ki Lee 2010b, p. 6). As a result, a history of *minjung* composed solely of resistance increasingly distances itself from the actual *minjung* who have existed throughout history. Even in *minjung* theology, the complexity of events in *minjung*'s history, which could not be reduced to the sole line of argument of the *minjung* movement, is overlooked. Thus, some *minjung* theologians, in particular Nam-Dong Suh, construct an evolutionary and teleological "line" of development that the suffering of the *minjung* coincides with the very movement of the conversion of force that moved history forward in the history of the *minjung* (e.g., Nam-Dong Suh 1983, pp. 225–26).

Therefore, could critical issues raised against the *minjung* theory of Korea in the 1970s and 80s truly be applied to Byung-Mu Ahn's theory on "*minjung*-event"? This paper aims to demonstrate that they could not. As is well known, unlike the majority of contemporary *minjung* theorists, the first generation of *minjung* theologians, especially Byung-Mu Ahn,

showed a strong aversion to conceptually defining who *minjung* were. However, this could not imply that Ahn thought understanding *minjung* was impossible. On the contrary, he endeavored to understand *minjung* more than anyone else (e.g., Ahn 2019, p. 221). Ahn was particularly able to develop his unique *minjung* theory by discovering the term "*ochlos*" (ὄχλος) in the Gospel of Mark. According to him, *ochlos* refers to the so-called "sinners" and the ritually impure in Jesus-era Palestine, the marginalized layers of society. While *ochlos* has commonly been translated as the crowd, the public, the masses, and so on, Ahn translated it as *minjung*. By doing so, he intended to provide a biblical and theological basis for the term *minjung* (Ahn [1975] 2013c, pp. 91–97; Ahn [1979] 2013a, pp. 49–64; Ahn [1981] 2013b, pp. 65–90), which, in the context of the 1970s Korean *minjung* movements and *minjung* projects, referred to "the majority of a society, the dominated class, the subject of historical development, and an entity with strong resistant and transformative potential" (In-Cheol Kang 2020, p. 219).

Based on what we have seen so far, Ahn's concept of "*ochlos*/*minjung*" seems to allude to the lower social strata, closely resembling a class-based entity or, collectively, the general populace of the lower class. However, Ahn denied such a reasoning. According to him, *ochlos*/*minjung* is not reduced to the proletariat, which in capitalism does not own the means of production. Furthermore, it is not reduced to a nation as the substance of nation-state or subjects of a democratic polity like "the people" or "citizens" (Ahn [1979] 2013a, p. 63). If "*ochlos*/*minjung*" is neither classes, sociological entities, nor political subjects, then are they the Beings (*das Seiende*) whose Being (*das Sein*) must be explained through which category?

> For us, the *minjung* appeared as an event. It means we encountered the *minjung* in the event. Encountering with the *minjung* demanded a Copernican revolution in our thinking. It forced us, so to speak, to adopt a new way of thinking. Therefore, this event did not merely bring us a new understanding, but it also prompted our conversion as theologians. (Ahn 1995, p. 40)

Ahn's answer to the question "Who are *minjung*?" can be summarized as "The *minjung* are an event." According to him, "*Minjung* can only be experienced and is not an object of intellectual understanding. We have clearly seen. We have experienced. We are witnessing to the *minjung*-event just as we have seen it" (Ahn 2019, pp. 220–21). He perceived *minjung* as an event itself, manifesting internal ruptures of existing (political, economic, and social) structures in dominance (structures articulated in dominance). For Ahn, the term *ochlos*/*minjung* was conceptualized as the "eventual subject" only when rediscovered in the event, specifically when his theology transformed into a theology of the event following "the Jeon Tae-il event".[7] Thus, he made it clear to *minjung* theologians including himself that *minjung* always appeared as an event, that is, that *minjung* is only encountered in and through events.

Ahn perceived events involving Jesus and the *ochlos*/*minjung* in Palestine 2000 years ago as one of the historical events in the "volcanic lava" of the *minjung* liberation movement (though, as a theologian, the Jesus-event symbolically held the most central position for Ahn himself). Here, *minjung* emerged as a unique category of subject, appearing as the crack in the universal substance or constitutive void of structure in dominance. Just as the "*ochlos*" can be understood in contrast to "*laos*" (λαος, meaning nation or citizens) and their true nature is discerned only through the Messiah event or the liberation event manifested in the Jesus-movement they initiated with Jesus, in contemporary *minjung* theology, *minjung* is also an "eventual subject". Their nature can be understood only in contrast to collective subjects/sociological entities like citizens, nation, ethnic groups, and proletariat.

Just as the *ochlos* cannot be understood without its distinction from *laos*, *minjung* cannot be grasped without juxtaposition to entities like citizens, nation, proletariat, and ethnicities. However, this opposition should be carefully grasped because *minjung* is not simply another entity paralleling these categories. Instead, it can be seen as "the Real" (Lacanian) that decodes or deconstructs these collective subjects or sociological entities when viewed through the lens of an event (Žižek 1999, p. 102). Borrowing terminology

from Heidegger, whom Ahn deeply engaged with, *minjung* can be said to be a "There-Being" (*Dasein*) that can pose a question of existence to those categorized as collective subjects or sociological entities.

Therefore, Ahn's *minjung*-event theory is a product of his struggle to escape both limitations of *minjung* theory in general and neo-orthodoxy theology by emphasizing the event in the debate with conventional social scientific views of *minjung* and concurrently by emphasizing *minjung* against existentialistic interpretations on the event. In terms of the general *minjung* theory, it highlights the presence (*Anwesen*) of *minjung* as an "event" that transcends the limits of cognition. In terms of the general event theory, by emphasizing the "*minjung* partisanship" of divine revelation, Ahn succeeded in presenting a dialectical theory that was both event oriented and *minjung* oriented, theological and sociological, transcendental and immanent.

To empirically justify his own perspective, Ahn confessed that he and other participants in the Korean democratization movement of the 1970s and 80s encountered the present Christ directly in *minjung*-events that took place on the fields of the movement. "Just as with the Jesus-event, Korean *minjung* theologians are experiencing the current Christ-event by participating in such *minjung*-events. *Minjung* theology is experiencing the Christ-event on the fields by participating in the *minjung*-events" (Ahn 1993b, p. 82). Based on this experience, Ahn defined *minjung* theology as "the work of theologically examining the *minjung*-event" (Ahn 2019, p. xix). He saw the task of *minjung* theology as clarifying not the *minjung*, but the "*minjung*-event". Why did Ahn emphasize the *minjung*-event rather than the *minjung* itself? It was because he believed that the presence of *minjung* could only be grasped within and through the event. He frequently used the term "*minjung*-event" to present the *minjung* as an entity consubstantial with the event. By emphasizing this synonymous relationship between *minjung* and the event and expressing *minjung* as an event, he distinguished himself not only from other *minjung* theologians, but from all *minjung* theorists of the 1970s and 80s. Then, what is the specific nature of the *minjung*-event for Ahn?

> We theologize the *minjung*-event. [...] The fact that over 40 students and workers have sacrificed their lives like this speaks eloquently of our current situation. We, who practice *minjung* theology, readily consider such students as part of the *minjung*. Workers also committed self-immolation. How can this be possible? Facing these facts, I had no choice but to make this confession: The *minjung* can transcend themselves. In everyday life, it's impossible to comprehend such actions. Yet, in our midst, events of self-transcendence keep occurring even among non-Christians. [...] I see this as the *minjung*-event. (Ahn 1999, pp. 179–82)

Here, the emphasis is on the fact that the *minjung*-event signifies the suffering and self-transcendent sacrifice of the *minjung*. However, Ahn did not solely perceive the *minjung*-event within the context of *minjung*'s suffering or their self-transcendent sacrifice.

> Of course, '*minjung*' is a collective concept. [...] However, it's not a static concept but a dynamic one. Being dynamic implies its defensive and militant nature. *Minjung* is partisan. Therefore, you can only see the *minjung* when looking with a partisan perspective. They are unconditionally generous and inclusive towards their side. But they are militant against adversaries. [...] This is inherently dynamic. That is what *minjung* is. We observed this aspect in the *minjung* movement. (Ahn 1993a, p. 229)

For Ahn, *minjung* revealed its presence only in the liberating struggle against power and violence. The *minjung*-event flowed continuously beneath the crust of history, like a volcanic lava, and erupted to the surface under specific historical conjuncture. This suggested that for him, the *minjung*-event was understood as an ontological event and a necessary law of history. According to a third-generation *minjung* theologian Jin-Ho Kim's explanation, "as long as there is the situation of exploitation that pits power against counter-power, the presence of *minjung* as a subject of practice which deconstructs the power-structures

is perpetual. It's just that in 'everyday life', it's latent in the dimension of individual or group-specific actions and does not get captured as an eventful substance. Therefore, while *minjung* aims to be embodied in a more effective formal political alliance, its presence is inherent in everydayness" (Jin-Ho Kim 1995, p. 84).

On the surface of historical perception, *minjung*-events seemed to be interrupted. Therefore, they could not be grasped by a historicist view of historical criticism. However, in terms of historical consciousness, there is a vast interconnectedness underneath, like volcanic lava. Ahn emphasized that *minjung* is concealed and hidden, pointing out that it appears alongside events that bring about historical turning points. More precisely, *minjung* can only be understood in and through such events. This was why he used the term "*minjung*-event", which seemed tautological, to express *minjung*.

## 4. Religious Significance of the *Minjung*-Event Theory: A Project of Profanation of the *Minjung* as a Sacralized Substance

"Events should be conceived of as sequences of occurrences that result in transformations of structures."

—Sewell (2005, p. 227)

The definition of "*minjung*" established in the Korean academic field in the 1970s was "a group of people fighting to dismantle the oppression and exploitation caused by political, economic, and social contradictions" or "a category of people who are marginalized not only economically but also politically and culturally" (In-Cheol Kang 2023a, p. 79). Among *minjung* theologians, Nam-Dong Suh clearly articulated this definition of *minjung*. According to him, "a tentative definition of *minjung* is the poor, oppressed, and estranged people" (Suh 1983, p. 227). Byung-Mu Ahn largely agreed with this definition of *minjung*. However, as discussed previously, his originality laid in perceiving *minjung* as "an eventful subject" created with the advent of a specific moment called an "event", "the situation that makes visible/legible what the 'official' situation had to 'repress'" (Žižek 1999, p. 130). Hence, he argued that the *minjung* could not be reduced to a class or a set of social attributes (income, education level, etc.). In Ahn's *minjung*-event theory, *minjung* is not a constantly observed group of people, but a phenomenon that unexpectedly emerges in the course of history, refracting the flow of the world and then, being presented as those who live ordinary, mundane, and simple lives, submerges back into the materiality of everyday life. As sociologist Hong Jung Kim pointed out, in fact, this conceptualization of the eventful *minjung* was similarly found in Western academic discussions about "the people" (Hong Jung Kim 2021, p. 131).

For example, French political philosopher Pierre Rosanvallon, noting that modern people are a fundamentally invisible group, calls the people erupting from historical events a "people-event" (*peuple-événement*) (Rosanvallon 1998, pp. 53–55), which is almost identical to Ahn's concept of the "*minjung*-event". To derive the expression of "people-event", Rosanvallon first explains that the people have two bodies. As he explains, this is because "the people" always refers to two distinct concepts. In political theory and constitutional law, "the people" denotes a unified agent in which sovereignty is invested. In social sciences, however, "the people" denotes a heterogeneous population, numbers unlimited, that can only be grasped as figures and statistical averages. Rosanvallon names the former people as "people-as-sovereign" and the latter as "people-as-society". According to him, the people-as-sovereign is well defined through apparatuses like "a popular vote, a parliament, or a house of representatives, which at once expresses and confirms the principal unity of the people." Thus, the people-as-sovereign is "dense and energized by the principle of unity that it both expresses and confirms." In contrast, the people-as-society "are evasive, abstract, and formless, pure number and seriality" (Rosanvallon 1998, pp. 35–55; Jonsson 2006, p. 55; 2008, p. 23).

In the "*Minjung* Project" of the 1970s and 1980s in Korea, which included *minjung* theology, the *minjung* represented two distinct concepts, just like the people in France. This meant that the *minjung*, like the people, embodied both the character of the *minjung*-

as-society and the *minjung*-as-sovereign. This was because the *minjung* was depicted as having both the majority aspect from the standpoint of the *minjung*-as-society and the historical subjectivity from the standpoint of the *minjung*-as-sovereign. First, the *minjung* as an "overarching concept that encompasses various notions such as class, ethnicity, and citizen" (Hyun-Chae Park [1984] 2006, p. 99) is "composed of various classes and strata, and the complexity of *minjung* composition can also encompass differences such as gender, race, and generation" (In-Cheol Kang 2020, p. 221). In this regard, the *minjung*, in terms of its majority, like the people-as-society, refers to a heterogeneous population that can only be grasped as a number. Naturally, because of this internal diversity and heterogeneity, which can only be captured by numbers, the *minjung*-as-society reveals the nature of the *minjung*'s subordination, their characters as objects of rule. At least in the age of Korea's developmental dictatorship, the heterogeneity and diversity of the *minjung*-as-society essentially formed prerequisites for *minjung*'s subjugation.

On the other hand, the *minjung* as historical subjectivity, similar to the people-as-sovereign, "may not be the 'sole agent' of history, but they have positioned themselves as one of the agents that co-create history amid conflicts with the ruling power. Furthermore, the *minjung* has often been considered the 'true agent' of history" (In-Cheol Kang 2020, p. 220). In this respect, the *minjung*, like the people, refers to a unified agent in whom sovereignty is invested. Naturally, the unity of the *minjung*, when combined with the principle of popular sovereignty, has transformed the *minjung* from being mere objects and absentees of history to active subjects and presences within it. Thus, the unity of the *minjung*-as-sovereign is a prerequisite for historical subjectivity of the *minjung*.

This contradiction inherent in the concept of the *minjung*, in the form of the *minjung*-as-society and the *minjung*-as-sovereign, led to "duality thesis of the *minjung*" throughout the length and breadth of Korean *minjung* theories in the 1970s and 1980s, which was divided between "a set of concepts referring to the 'deprived' on the one hand, and the 'resistant political subject' on the other" (In-Cheol Kang 2023a, p. 53). In particular, *minjung* theologians have formalized this "duality thesis of the *minjung*" into the logic of "the *minjung* as both 'bearers of suffering' and 'subjects of history'" (Jin-Ho Kim 2000, p. 92). Borrowing words from Byung-Mu Ahn, the *minjung* discovered through the *minjung*-event "were the bearers of history, even though they were in a peripheral position in human society" (Ahn 1993a, p. 332). How, then, did Ahn resolve the contradiction between these two different conceptions of the *minjung*? Of course, it was through an alternative concept of "*minjung*-event".

Interestingly, Rosanvallon also argued that the two distinct bodies of the people, the people-as-sovereign and the people-as-society, were merged into one through the people-as-event called the "French Revolution" (1789). This also has important implications for understanding the emergence of the concept of the *minjung*-event in *minjung* theology. According to Rosanvallon, the French Revolution sharpened the contrast between the formal act of voting and the more organic process of social identification. However, it also blurred clear lines of this distinction as unfolding events temporarily reconciled abstract societal concepts. Certain monumental moments are essential for these inherent tensions to appear reconciled. When the people were celebrated in 1789, a transformative moment occurred: the "people" came into the limelight, emerging from ambiguity to become an undeniable force. At the same time, the visible and the symbolic converged, allowing concept materialization in action. This manifestation of the people as an event, the occurrence of the "people-event", temporarily resolved contradictions found in the two opposing representations of the people—the people-as-sovereign and the people-as-society (Rosanvallon 1998, pp. 34–41).

As Stefan Jonsson, an ethnic studies researcher from Sweden, states, "the heterogeneous, many-headed people is unified into one political agent by the historical circumstances themselves, in order to institute 'the people' as a principle of sovereignty" (Jonsson 2008, p. 23; Jonsson 2006, p. 55). When synthesizing descriptions of Rosanvallon and Jonsson, it becomes evident that people-as-sovereign and the people-as-society merges

into the concept of the people-as-event, namely the emergence of a third concept of the people called "people-event" and the inherent contradiction in the modern notion of people reflecting the foundational aporia of democracy could be temporarily resolved (Rosanvallon 1998, p. 375). Similarly, the two concepts of *minjung* that appeared in Korean *minjung* theories including *minjung* theology in the 1970s and 80s—the *minjung* as a majority and the *minjung* as a historical subjectivity—are fused into the concept of the *minjung*-as-event in Byung-Mu Ahn's *minjung*-event theory. What distinguishes Ahn from other *minjung* theorists active in the 1970s and 80s is his attempt to reconcile the *minjung* as a majority with the *minjung* as a historical subjectivity through the concept of a "*minjung*-event."

> Some theologians came to recognize structural evil through the military regime. Those who were generally politically liberal began to physically experience this tightening grip of structural evil. [. . .] In the process of resistance, they realized that what the Bible calls to 'Satan' or 'the devil' is none other than this structural evil of power. In the process, they met the *minjung*, and only after encountering the *minjung* did they experience that the *minjung* are the bearers of history, who are not only thoroughly robbed and oppressed by this structural evil, but also do not succumb to it in the end. They came to realize that the *minjung* are truly the source of life and the subjects of history. The encounter with the *minjung* was a profound event, and this event led to a series of events that liberated the theologians from traditional theology. (Ahn 1993b, pp. 219–20)

In the quote above, Ahn speaks of his post-evental realization that the *minjung*, after encountering the "*minjung*-event", are both victims of structural evil—which presupposes the *minjung* as the social majority—and at the same time the subjects of history—which presupposes the *minjung* as the political sovereign. In Ahn's *minjung* theology, the *minjung*-event is "a moment when a given order of domination and a given regime of hierarchy are radically challenged by the emergence of a political subject", the *minjung* (Chambers 2013, p. 8). In other words, the *minjung* does not exist before an event, but comes into a being through the *minjung*-event, by creating a polemical common ground to expose structural evil and to claim human and civil rights, the right to work, and so on. For, in Rancière's terms, the *minjung* has no existence as a real part of the society before the wrong—that is, the *minjung*-event—that its name exposes (Rancière 1999, p. 39). The quote powerfully underscores that the *minjung* "undoes a given order, does not and cannot exist prior to its surprising and unpredictable appearance on the stage" of the *minjung*-event (Chambers 2013, p. 8). Therefore, the emergence of the *minjung*, the evental subject, is always untimely in the sense that the *minjung* is intelligible, visible, and tangible as such only after the moment of the *minjung*-event. In short, the *minjung*-event makes the *minjung* possible as an evental subject. It retrospectively and retroactively produces the eventual subject—that is, *minjung*—that appears to precede the event. The *minjung*-event is that original moment of disagreement that brings about the very existence of the *minjung* in the first place.

Both the *minjung*-as-society and the *minjung*-as-sovereign are products of discursive re-presentations that erase all differences that could not be erased in reality and imagine the *minjung* as a homogeneous collective entity, whether in the sociological or political sense.[8] Including Nam-Dong Suh's the *minjung* messianism, almost all *minjung* theories in Korea during the 1970s and 1980s overlooked the fact that the *minjung* were a product of re-presentation. Most *minjung* theorists understood that the *minjung* existed prior to the *minjung*-event as main actors in historical development with strong transformative potential. Thus, they thought that the *minjung*-event occurred as a result of the *minjung*'s resistance to power under a specific conjunctural conditions, as if the *minjung*-event arose due to resistant practices of the pre-existing *minjung*.

Ahn's *minjung*-event theory offers a notable departure from such conventional understanding. For Ahn, the *minjung* did not actually exist before the *minjung*-event. This gives rise to an alternative comprehension of the *minjung*. To paraphrase a political theorist Sofia Näsström's descriptions of democratic revolution, the *minjung*-event does not pass

through the experience of the *minjung*. This is because the minjung-event is synonymous with the minjung itself (Näsström 2006, pp. 334–35). Just as a democratic revolution exists only in the moment of its enactment, the *minjung* also exists solely at the moment of *minjung*-event's enactment. In this sense, in Ahn's aforementioned quote, "for us, the *minjung* appeared as an event", the term "appeared" does not refer to an a priori substance, nor does it signify a robust ontological consistency of the *minjung*. By equating the *minjung* with an event, Ahn avoided both pitfalls. To borrow a phrase from German writer Hito Steyerl, the *minjung* as an event exists as an enactment rather than a fixed attribute because it represents a dynamic process that evades predictable patterns, emerging "in that sudden blink of the eye that is not covered by anything" (Steyerl 2012). The *minjung* thus surfaces as an anomaly, contrasting with predictable patterns or those aspects beyond algorithmic predictability (Arditi 2014, p. 100).

> An event comes as a shock that transcends the framework of logic. Every event is unique. Just as a positive and a negative collide to create a spark, the event is that spark. However, being singular does not mean that it ends once and for all; just as the polarities of the universe continually ignite sparks at random points, an event is not isolated but has an intermittent-yet-successive nature. In the 70s and 80s, a series of events unimaginable to us occurred within the *minjung*. A prime example of this was the series of self-immolations. Starting with the self-immolation of a young worker named Jeon Tae-il, it became a series of events involving many students and citizens. Then we became aware of the connection between one event and another. (Ahn 1995, p. 41)

Thus, for Ahn, the *minjung*, borrowing In-Cheol Kang's expression, is an "existence that arrives through an ahistorical sudden event", more specifically, an "existence of contradictions and cracks that momentarily exposes the inherent oppressiveness and violence of the established order through 'unexpected events' like sporadic resistance or industrial accidents, only to swiftly vanish afterward" (In-Cheol Kang 2023a, p. 35; cf. Jinkyung Yi 2010, p. 103). In this way, Ahn's *minjung*-event theory distinguishes it from other *minjung* theories, preventing *minjung*'s substantialization. This is because the *minjung* can only emerge within history as an event that is both discontinuous and continuous.

Furthermore, if one agrees with a cultural anthropologist Marshall Sahlins's insight that "an event is indeed a happening of significance, and as significance it is dependent on the structure for its existence and effect" (Sahlins 1985, p. 153) and a historical sociologist William Sewell's observation that "the reason that events constitute what historians call 'turning points' is that they somehow change the structures that govern human conduct" (Sewell 2005, p. 218), then the occurrence of the event and the concomitant emergence of the *minjung* cannot be separated from the social structure. If the occurrence of the event cannot be fully understood as external to the social structure, then the *minjung* as the evental subject must also be understood as the posited others of the social structure, embodying the possibility and impossibility of the structure. Therefore, the *minjung*, which only emerges through events that unpredictably change social relations beyond their causal connections and subsequent gradual changes, can be seen as the horizon of the structure. In other words, although the *minjung* always escapes the grasp of the structure, it is a critical point in the structure that can only be represented within the structure (Wang 2008, p. 492).

From the perspective of the project of profanation, the significance of *minjung*-event theory, which presents the *minjung* as an eventful subject and an agency that is implied by the existence of structure, is that it not only avoids reiterating limitations of *minjung* theories that sacralize the *minjung* as the messianic beings that replace Jesus Christ, but also, crucially, allows for a critical re-examination of the idea of the "subject of history" as a teleological constant in the *minjung* theology. The idea of the minjung as the "subject of history" presupposes a unified identity of the *minjung*, whether spontaneous or acquired as the result of a process of formation and coming to consciousness, which is guaranteed by an already teleologically determined state. However, according to Ahn, the *minjung* is not a pre-existing or an a priori given subject prior to the event. Therefore, it must be emphasized

that the *minjung*, which only emerged through the *minjung*-event, never coincided with itself before the *minjung*-event. Of course, the fact that the *minjung* can be ontologically divided into pre- and post-event does not mean that groups identified as the *minjung* never present themselves or act as subjects in history. Rather, it emphasizes that the occurrence of a *minjung*-event, the emergence of the *minjung* as an evental subject, is always tied to a conjuncture, whether permanent or not, and exists only within its limits (Balibar 1988, p. 46; 1994, p. 147).

## 5. Conclusions: Contemporariness of the *Minjung*-Event Theory

> "The subjects a political wrong set in motion are not entities to whom such and such has happened by accident, but subjects whose very existence is the mode of manifestation of the wrong."
>
> —Rancière (1999, p. 39)

For Ahn, the *minjung*-event is a phenomenon of "the sacred" that arises in the specific historical and social context of Korean society, as the *minjung*-event fulfills conditions of both "deviating from the existing classification system and being unexpected and being considered perfect in the perspective of an ideal norm" (Suk Man Jang 2008, p. 372; Anttonen 2000a, p. 203). This is because, in the *minjung*-event theory, an event embodies the essential element of unpredictability. It cannot be reduced to any social norm or causality. Thus, it has effects not encompassed by its conditions. It is captured as a specific "rupture" moment. In this sense, by presenting the *minjung* as an event that is essentially unnameable, invisible, and unpredictable—a phenomenon of the sacred—the *minjung*-event theory can be seen as a theological re-appropriation of the sacred. The *minjung*-event is not only a sacred event that cannot be reduced to (or deduced from, generated by) a (previous) situation, but also involves a subjectivity that can only be constituted through participation in this sacred event. The engaged "subjective perspective" on the sacred event is part of the event itself and the *minjung*-event is such a sacred event as it is interpreted. Such an understanding of the event demonstrates the interpretation of contemporary religious studies, particularly the cognitive science of religion, that "the sacred is first and foremost a cognitive category, the representations of which are culture-dependent" (Anttonen 2000b, p. 277).

Nevertheless, it is important to note that when equating the *minjung* with the event, a phenomenon of the sacred, Ahn's *minjung*-event theory identifies the event not merely as a rupturing other to the social structure, but as simultaneously forming part of it in some way. This is because the agency revealed through events consists in the structural inconsistencies that open structural gaps available to social actors. For example, Ahn, who has argued that the *minjung* is an event, declares that "Jesus is an event! God is an event, too!" (Ahn 2019, p. 18). In Ahn's theological system, the *minjung*, Jesus, and God are all events, forming a serialization of sense. Ahn further developed his understanding of God in relation to the thesis of "The God of *Minjung*". He argues that "there are various descriptions of god in the Bible, but essentially they point to a god that causes an event to occur" (Ahn 2019, p. 107). The point is that these events only happen in history, or, more precisely, within the social structure. Like the *minjung* as an event, God as an event is not only a constitutive outside of the world, conceptually distinct from the world or life, but also ontologically inseparable from it. Indeed, Ahn ventures so far as to equate God to an event with contradictions of the world itself.

> God does not exist as a responder or problem-solver in another realm beyond this world where humans are groaning and crying out in the middle of life and events. God exists right here in the midst of the crying out. Therefore, the God of the Bible is not the answer to a riddle of the universe or life but is the question from the conflicts and contradictions of life itself. The God of the Bible is not perfection or harmony, but conflict and contradiction. Rather than harmonizing the world, God causes problems. God keeps making events happen. And God takes contradiction to extremes. (Ahn 2019, p. 107)

Ahn's position on *minjung*-event theory is to identify the constitutive void in the structure, the social structural contradictions, and not only the *minjung*, but also the God of *Minjung* within the framework of events. Such an identification already "ontologizes" the *minjung*, albeit in a purely negative way—that is, it turns the *minjung* into an entity consubstantial with the structure, an entity that belongs to the order of what is necessary and a priori ("no structure without the *minjung*"). *Minjung* is the negative gesture of breaking out of constraints of being that opens up the space of possible subjectivation (Žižek 1999, p. 159). In this respect, *minjung* theology, "the work of theologically examining the *minjung*-event" moves beyond the secularization of the Messiah into the *Minjung*. This is because *minjung* theology "marches together with the minjung event but cannot ever be stagnant within a certain form" (Ahn 2019, p. xix). Therefore, *minjung* theology is a dialectical project of desacralization and re-sacralization that seeks to rewrite "the religious" through events, while at the same time profaning the *minjung* themselves, who have been sacralized as the messianic beings, into an event once again.

**Funding:** This research received no external funding.

**Institutional Review Board Statement:** Not applicable.

**Informed Consent Statement:** Not applicable.

**Data Availability Statement:** Not applicable.

**Conflicts of Interest:** The author declares no conflict of interest.

## Notes

[1]  *Minjung* theology is a (South) Korean theological discourse that has been developing from the mid-1970s to the present day, taking the Korean term '*minjung*" for political subjects, the equivalent of the people in English or *das Volk* in German, as the subject and object of theology. For a history of *minjung* theology, its central figures, and key concepts, see Kim and Kim (2023).

[2]  *Minjung* theology has been well understood in the international theological community as a type of "liberation theology" or "political theology" specifically tailored to the Korean context from an early stage. For instance, Jürgen Moltmann, one of the leading German Protestant systematic theologians of the second half of the 20th century, reminisced in depth on his encounter with *minjung* theology in the 1970s in his theological autobiography. Therein, he characterized *minjung* theology as "the first liberation theology to come from Asia, with critical questions put to the First World" (Moltmann 2000, p. 250). Furthermore, he assessed that *minjung* theology, which "links Jesus' gospel of the poor (*ochlos*) with […] native popular and resistance traditions," was "not a theology that has been made culturally indigenous, like 'yellow theology' before it", but "a contextual theology of the suffering people in Korea" as well "Korea's first political theology" (Moltmann 2000, p. 252). Following Moltmann, Volker Küster, who has been actively contributing to the widespread recognition of *minjung* theology in the current German-speaking theological community, also asserts in his monograph that "*minjung* theology has been categorized as 'Korean-style liberation theology'" (Küster 2010, p. xvi). In a recent collection of research papers on *minjung* theology, which he co-edited with a Korean *minjung* theologian Jin Kwan Kwon, he maintains the longstanding stance in a his own essay related to Nam-Dong Suh recognized as a co-founder of *minjung* theology with Byung-Mu Ahn, characterizing *minjung* theology as "the South Korean brand of liberation theology" (Küster 2018, p. 25).

[3]  Another first-generation *minjung* theologian, Nam-Dong Suh, also views secularization as a counter-theology that opposes both existing "theology of domination" integrated into the ruling (dominant) ideology and the "theology of the head" (speculative/deductive) in favor of a "theology of the body" (practical/inductive). Therefore, along with Ahn, Suh can also be seen as actively practicing "secularization" to break free from the authority and dominance of Western Christianity (Nam-Dong Suh 1983, p. 352).

[4]  In religious studies and sociology in general, the opposite of "secular" implied by the concept of secularization is understood to be "religious" rather than "sacred". Instead, the opposite of "sacred" is identified as "profane". Therefore, it is necessary to distinguish the usage of the concept of secularization more strictly between Weberian (a change in the overall world order from "religious" to "secular") and Durkheimian (a process of de-centralization and diversification of the "sacred"). For difference between Weberian and Durkheimian approaches to secularization, see Boskoff and Becker (1957, pp. 133–85) and Nisbet (1966, pp. 243–51).

[5]  In theology, where an omnipotent God secures its authority by performing miracles that transcend the ordinary, and in jurisprudence, where the sovereign reveals its dominion by determining the state of exception—a state in which the legal order itself is suspended and exists outside of valid legal systems—Schmitt reveals the essence of his political theology by arguing for a structural equivalence between the two. Ultimately, he reintroduces transcendental categories into political thought through

secularized theological concepts, deriving from theology notions like the sovereign (God), the state of exception (miracle), and the decision (divine intervention).

6   The "New Style Phenomenology of Religion", proposed by the Dutch scholar of religion Jacques Waardenburg as an alternative paradigm to classical phenomenology of religion within the Swiss–Dutch school, was established through differentiation from the classical approach. Its most significant distinction is perceiving religion not as a pre-existing "reality" imbued with an intrinsic religious attribute of "sacrality", but as a specific "phenomenon" stemming from "various patterns of behavior and interpretations concerning reality" (Waardenburg 2001, p. 455). Thus, in the new style phenomenology of religion, there is no such thing as a fixed religious phenomenon. It acknowledges that these "phenomena" can change due to the actions of the religious actors involved, and places greater emphasis on the social structures and historical contexts in which the categories like "the sacred" or "sacrality" are constructed. In this regard, Waardenburg contends that in order to strengthen the epistemological foundation of the phenomenology of religion, the conception of "intentionality", which can be defined as "research such subjective interpretations of meaning as objectively as possible", should be reintroduced to religious studies (Waardenburg 2001, p. 454; Waardenburg 1978, p. 116ff). Therefore, in order to discover the "surplus value" (*Überschusswert*) in religious phenomena that holds meaning for believers, one must first define religion as the "reality of meaning" (*Sinnwirklichkeit*). This is because, from the perspective of a religious scholar, religious phenomena are considered to be more than the factual data found in empirical studies, but rather the subjective meanings that believers themselves attach to the ultimate values contained in various religious expressions (Waardenburg 1978, pp. 113–17). Therefore, the new-style phenomenology of religion can be defined as a discipline that "empirically researches the meaning and interpretation of religious appearances as human phenomena" (Waardenburg 2001, p. 449). Waardenburg argues that this new style phenomenology of religion should be concretized by attempting a new approach to the "social reality", specifically the traditions, society, culture, and ideologies that religious actors are directly encountering here and now, with the intentionality of religious phenomena as its axis. Based on the theory of the new-style phenomenology of religion, this paper analyzes a historically specific Korean religious phenomenon called "*minjung* theology" as one of the contemporary living religions.

7   Jeon Tae-il (28 September 1948~13 November 1970) was a Korean sewing worker and workers' rights activist who committed suicide by self-immolation at the age of 22 in protest of poor working conditions of Korean factories during the fourth Republic era. His death brought attention to the substandard labor conditions and triggered the formation of labor union movement in Korea. In the 1970s, the decisive catalyst for the emergence of a unique indigenous theological discourse called "*minjung* theology" in Korea was also the Jeon Tae-il event.

8   The *minjung* as the subjugated, which corresponds to the *minjung*-as-society, does not pre-exist the process of governing towards "a population as wealth, a population as work force, a population balanced between growth and resources, a population with special phenomena and unique variables". In the same way, the *minjung* as historical subjectivity, which corresponds to the *minjung*-as-sovereign, cannot exist without the process of reproduction through apparatuses that both express and confirm the basic unity of the people, such as "a popular vote, a parliament, or a house of representatives". Just as the *minjung* as a social majority, which presupposes the *minjung* as the subjugated, does not exist prior to the operation of technology of government such as demographics, the *minjung* as sovereign, which presupposes the *minjung* as historical subjectivity, does not exist prior to processes of political re-presentation such as voting. This means that the *minjung*, represented as the majority, the subjugated, the sovereign, and the subjects of history, are a product of re-presentation as a process that constitutes the thing or object itself, not as an already established substance or the thing-in-itself independent of the process of re-presentation.

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
