# Peer review of "Minjung Theology as a Project of Profanation: Focusing on the Minjung-Event Theory of Byung-Mu Ahn"

_religions, doi:10.3390/rel14111395_

Round 1

Reviewer 1 Report

Comments and Suggestions for Authors

I think this article is interesting and worth publishing. I have some issues though, coming from a different background. It takes for granted some presuppositions that are presented in a generalised/essentialist way, that could raise a lot of discussion. Eg. note 4 about the terms secular/religious, sacred/profane, or the idea attributed to all "modern religious studies" about the sacred being first and foremost a cognitive category (lines 720-721, with a reference to Anttonen). This is a dabatable argument/position, and there are certainly more than one ways to study religion. One reference is not enough to support this as a doctrine of religious studies methodology overall. Perhaps one way to remedy this would be to refer to a particular way of viewing religious studies (the same way it is done when naming the "new style phenomenology of religion"). Where I come from (from a semi-Western background) there would be much reservation about using the terms secularisation or profanation in an exclusively positive manner, also as a desideratum. Taking this aside, the article is interesting and challenging in different ways. It is a bit difficult to read, given the many theoretical presupositions, but it is certainly worth reading. Line 82, it is not very clear to what the word it (recognizes it) refers (the process of secularization or religion). Line 105, in the end of the first sentence, it would be good to announce that the distinction between secularization and profanation will be discussed in what follows, because the reader feels an explanation is needed. Similarly, note 6 should come a bit earlier in the text, as the reader maybe scratching his/her head. The end of paragraph 4, could also be repeated in a way in the beginning of the paragraph. It is worth clarifying (or even repeating) how minjun is being sacralized (and the connection between substantialization and sacralization), so that profanation can take place. On another issue, the terminology based on ochlos and laos is a bit out of tone, but I understand this is how it has been used by Ahn. Still, ochlos has a very negative connotation in Greek, it would mean "mob", while as laos a rather positive one, meaning "the people" (distinct from nation). Another issue is the example of self immolation as a model for the minjung event, without any further commenting or challenging (this would not be my cup of tea). Despite all observations, I do think the article is much above average, in fact quite original, and could initiate a discussion.

Comments on the Quality of English Language

Not being a native English speaker, I just noticed some issues in the use of the definite and indefinite article (eg. line 159, is it an Italian philosopher or the Italian philosopher G. Agamben) and also the variations a and an, in more than one cases. Still, I am not the right person to comment on that. 

Author Response

1. Answer to “Comments and Suggestions for Authors”

First of all, I would like to thank you for your good review and incisive assessment of my paper. In particular, I take your point about "some presuppositions that are presented in a generalised/essentialist way" and have revised the states in a few problematic places. 

As you pointed out, I have also revised words and sentences on lines 82, 105, 163, and 722-724 to make them clearer. 

As for some of the theoretical issues you raise, I will give them more careful consideration and incorporate them into future work. Thank you very much.

Taking your point, I've marked the parts I've modified in blue. I hope you can see that.

2. Answer to Comments on the Quality of English Language

The line 159 you pointed out is correct to use the definite article 'the', so I have fixed it. Thank you for your precise point. I will consult with the editors of the journal Religions for more thorough revisions. Thank you very much.

Reviewer 2 Report

Comments and Suggestions for Authors

I congratulate the author on this fine piece of work. The works and scholars engaged are wide-ranging and excellent. Future work could surely expand the conversation further. Extending this conversation could be the work of a lifetime. Well done!

Author Response

I sincerely appreciate your generous review. As you suggest, I will endeavor to further develop many of the issues I have raised in this paper in future work. Thank you very much. 

Reviewer 3 Report

Comments and Suggestions for Authors

The paper reinterprets minjung theology as a project based on the conception of secularization and profanation influenced by the political philosopher Agamben. The claim is this approach and the various social, political developments in Korea sets minjung theology apart from the liberation theology of South America. In order to answer this the author provided a theoretical examination.  

The paper is divided into four sections. The first two discusses the sociological theories of Weber and Durkheim as then understood within the work of Agamben. This is followed by an introduction of Byung-Mu’s concept of minjung theology understood as minjung-event. The biblical grounding for this approach is from the Greek New Testament term ochlos as used in the Gospel of Mark. The fourth section attempts to describe how Byung-Mu’s description of the minjung-event overcomes some of theoretical limitations of the 1970’s and 1980’s. The conclusion tries to show the de-sacralization and re-sacralization process within minjung theology through the lens of minjung-event.

It is not clear to this reviewer that the secularization process as described by Weber, Durkheim and Agamben is necessary in understanding the minjung-event as described by Byung-Mu. It seems to make minjung theology more confusing to understand. The question is –did secularization occur in Korean society? Or, is secularization a phenomenon that is unique to Western countries? Is the deep religious landscape (shamanism, Buddhism, Confucianism, Christianity) of Korea immune from the secularization process? These questions may help make the paper stronger and provoke more interest. Also, what that is missing from this description minjung theology is the concept of han which also was an integral part of the creation and development of minjung theology.

Comments on the Quality of English Language

The formatting of the paper needs attention. There are letters that are separated from words that do not have a dash. It may they are just hanging separately within the sentence. This is a distraction in reading the text. Please re-format the text. Also, there are misspellings of words and run-on sentences that distracts from the text. 

Author Response

1. Answer to “Comments and Suggestions for Authors”

First of all, I would like to thank you for your good review and incisive assessment of my paper. 

Regarding your critique that “It is not clear to this reviewer that the secularization process as described by Weber, Durkheim and Agamben is necessary in understanding the minjung-event as described by Byung-Mu Ahn,” my response is as follows.

Firstly, as I clarified in the abstract and introduction, this paper was written with the understanding that “the relationship between minjung theology and the process of social change called secularization, or the theoretical and practical projects based on such a processes of social change, is complex and hence requires more detailed discussion.” Such an critical mind arose from the limitations of existing studies on the relationship between minjung theology and secularization that were submitted to this very journal Religions in which I submitted my paper. I would appreciate it if you could review the context in which this paper was written more carefully.

Furthermore, I fully agree with your observation that the concept of 'Han' is essential in minjung theology. 'Han' is a concept from Nam-Dong Sug, one of the co-founders of minjung theology. Since my paper mainly deals with the theory of another co-founder of minjung theology, Byung-Mu Ahn, I did not address the concept of 'Han' in this paper. I will be sure to reflect on this in my future research. Thank you for your valuable feedback.

2. Answer to Comments on the Quality of English Language

Thank you for your precise point. I will consult with the editors of the journal Religions for more thorough revisions. Thank you very much.

Round 2

Reviewer 3 Report

Comments and Suggestions for Authors

The revisions added to the article is appropriate and clarifies previous concerns.